# Metformin’s Mechanism of Action Is Stimulation of the Biosynthesis of the Natural Cyclic AMP Antagonist Prostaglandylinositol Cyclic Phosphate (Cyclic PIP) [note 1]

**DOI:** 10.3390/ijms23042200

**Published:** 2022-02-16

**Authors:** Heinrich K. Wasner

**Affiliations:** BioReg Biopharm, Technology Innovation Laboratory, University of Illinois at Chicago, 2242 West Harrison St., Ste. 201, Chicago, IL 60612, USA; hwasner@bioregbiopharm.com

**Keywords:** biguanides, cyclic AMP antagonist, cyclic PIP, mechanism of insulin action, metformin, prostaglandylinositol cyclic phosphate, protein tyrosine kinase, protein tyrosine phosphatase, protein serine/threonine phosphatase

## Abstract

Metformin is the leading drug for treating type 2 diabetics, but the mechanism of action of metformin, despite some suggested mechanisms such as the activation of the AMP-kinase, is largely unknown. Among its many positive effects are the reduction of blood glucose levels, the inhibition of cyclic AMP synthesis, gluconeogenesis and an increase in sensitivity to insulin. Recent studies have described the natural antagonist of cyclic AMP, prostaglandylinositol cyclic phosphate. Synthesis of cyclic PIP is stimulated in all organs by hormones such as insulin and also by drugs such as metformin. Its primary action is to trigger the dephosphorylation of proteins/enzymes, phosphorylated on serine/threonine residues. Cyclic PIP triggers many of the regulations requested by insulin. The parallels between the beneficial effects of metformin and the regulations triggered by cyclic PIP suggest that the mechanism of action of this key drug may well be explained by its stimulation of the synthesis of cyclic PIP.

## 1. Introduction

Type 1 diabetes results from the destruction of pancreatic β-cells by an autoimmune reaction leading to the inability to synthesize the hormone insulin. Type 1 diabetes is best treated by daily injections of insulin, substituting for the lack of the body’s own synthesis. Treatment is controlled by measuring the blood glucose levels as an indicator of the sufficiency of insulin. Additionally, the amount of glycated hemoglobin A1c is determined, and this is an indicator of the longtime efficacy of this treatment [1].

Around 90% of diabetics, however, have type 2 diabetes, which is connected at the onset not with a deficiency of insulin but with a decreasing responsiveness of all cells of the body to insulin [2,3]. This means the transduction of the insulin signal into all cells gradually decreases. This causes a decrease in the activation of blood glucose uptake by muscle and other cells and a decrease in the ability of pancreatic β-cells to switch off insulin secretion. This leads to elevated blood glucose levels and also blood insulin levels at the onset of the disease. Insulin is a premier anabolic hormone of the human body and its diminishing signal will lead to serious deficiency in recovery and restoration of many functions of a body. The loss of this anabolic function is most likely the main reason why diabetics develop so many secondary, life-threatening health issues. Insulin, along with glucagon, are essential for keeping the blood glucose level in balance.

Many patients with type 2 diabetes are overweight and viewed as being uncontrolled eaters. Present thinking assumes that this eating behavior causes the development of diabetes of type 2. However, there are also lean patients with type 2 diabetes [4] and, furthermore, there are lifelong overweight people who do not develop type 2 diabetes [5]. One has to be aware that the gradually decreasing effectiveness of insulin causes the dose-response curves of insulin action to shift more and more to the right. Additionally, one cannot expect that the dose-response curves of all the different cell-types and organs of a body shift in an identical fashion, just as this shift will differ in different people. One possible scenario, the dose-response curve of pancreatic β-cells is shifted the furthest to the right. Since insulin shuts off its own secretion, the diminishing response to insulin means that there is prolonged secretion, leading to increased blood insulin levels. However, the dose-response curves of most other cell types, e.g., muscle, liver and fat cells, are shifted less far and respond better to secreted insulin. This results in a reasonable uptake of glucose into various cell types and can lead to decreases in blood glucose levels below basal values, making a patient feel hungry shortly after having eaten. This patient will be susceptible to weight gain. In a second scenario the peripheral tissues of a body, such as the muscle and fat cells, may become less responsive to insulin while pancreatic β-cells remain more responsive. In this case, the increased blood glucose levels will continuously stimulate the secretion of insulin, but not sufficiently to reduce elevated blood glucose levels, depending on how far the insulin resistance has proceeded. The questions that need to be answered are: What are the final reasons for this shift and what can be done in order to prevent this shift? Since insulin itself is not changed, the reason should be found at the receptor level.

In recent years, an antagonist to cyclic AMP, prostaglandylinositol cyclic phosphate (cyclic PIP), was characterized [6,7,8,9]. Its dominant action is to trigger the dephosphorylation of phospho-serine/threonine proteins, which were phosphorylated by the action of cyclic AMP. The biguanides, metformin, buformin, and phenformin, trigger activation of cyclic PIP synthesis in the liver of rodents [10]. This raises the question of whether this activation of cyclic PIP synthase can explain the mechanism of action of the biguanide metformin, a common therapeutic for humans with type 2 diabetes.

## 2. The Biguanide Metformin

Metformin was identified in 1922 and was used as an oral antidiabetic drug beginning in the 1950’s. In the 1970’s, increased lactate levels after biguanide treatments, causing lactic acidosis, led to the decision to take the biguanides buformin and phenformin off the market. Metformin has remained available. Its beneficial effects for treating type 2 diabetic patients are accepted worldwide. It is the 4th most frequently prescribed drug in the USA today. Metformin is not only beneficial for type 2 diabetics. It is beneficial also for type 1 diabetics [11], it may help various cancer patients [12], and it can improve aging problems [13]. A proposed mechanism of action of metformin is that it inhibits the respiratory chain in the mitochondria leading to increased AMP levels. These activate the AMP kinase, leading to most of the needed effects [14]. Nonetheless, most discussions on this topic conclude that the molecular mechanism of metformin’s action is “complicated and not completely understood” [15,16]. However, in studies of well-understood metabolic pathways, actions of metformin have been characterized. It increases sensitivity to insulin, decreases elevated blood glucose levels, and inhibits gluconeogenesis and the synthesis of cyclic AMP [17].

## 3. The Natural Cyclic AMP Antagonist Prostaglandylinositol Cyclic Phosphate

The search for a counter-player to cyclic AMP started in the laboratory of the late Earl Sutherland. Insulin and noradrenalin (α-receptor action) stimulate the synthesis of cyclic PIP dose-dependently. Cyclic PIP levels increase several-fold, reaching a maximum after 1 to 3 min, then decline reaching basal levels after 15 min. It was isolated from all rat organs (brain, heart, intestine, liver, lung, muscle, spleen, testis), from pig liver and brewers’ yeast [18]. Its synthesis was characterized in monkey liver [19] and in the slime mold *Dictyostelium discoideum* [6]. Cyclic PIP is composed of prostaglandin E (PGE), *myo*-inositol and one phosphate. The inositol (1:2-cyclic) phosphate is bound by its C4 hydroxyl group to the C15 hydroxyl group of the PGE as determined by degradation experiments of cyclic PIP and mass spectrometry (Figure 1) [6,7,18]. It is biosynthesized [20] from PGE and an activated inositol cyclic phosphate, which is suggested to be guanosine diphospho-4′-*myo*-inositol (1′:2′-cyclic)-phosphate (Gypakis et al., manuscript to be submitted). Cyclic PIP synthase is found predominantly in the light microsomal fraction. It is active in a protein tyrosine phosphorylated form [21] and inactive in a protein serine/threonine phosphorylated form [10] (Figure 2). Basal cyclic PIP synthase activity is increased up to 3-fold by fluoride and up to 10-fold by ATP (Table 1). Initially it was assumed that the ATP is needed to drive the synthesis of the ether bond, but it is now clear that the ATP is solely needed to activate cyclic PIP synthase by protein tyrosine phosphorylation and the activated inositol cyclic phosphate contains the necessary energy for the synthesis of cyclic PIP [21].

The primary regulatory properties of cyclic PIP are the dose-dependent, 7-fold activation of protein serine/threonine phosphatase (P-ser/thr-P) holoenzyme and the dose-dependent inhibition of cyclic AMP-dependent protein kinase (PKA) [6]. Thus, the equilibrium between the phospho- and de-phospho-form of an interconvertible enzyme is regulated by the two enzymes PKA and P-ser/thr-P holoenzyme and the two regulators cyclic AMP and cyclic PIP [6]. For example, insulin stimulates the synthesis of cyclic PIP, which inhibits the adenylate cyclase and PKA, and activates the P-ser/thr-P holoenzyme. In this way a rapid shift of an interconvertible enzyme to its dephosphorylated form is obtained, since new phosphorylation of enzymes is shut off and existing phosphoproteins are dephosphorylated. It is essential that both enzymes, PKA and P-ser/thr-P holoenzyme, are regulated at the same time. For instance, if only the protein phosphatase is activated and the kinase is not inactivated, futile cycling would result, but no change in the regulation of metabolism would be reached. Previously, cyclic AMP has been shown to activate but not inhibit protein kinase A. Thus, one can assume that catabolism by cyclic AMP is switched on by phosphorylation of interconvertible enzymes. Alternatively, anabolism must then be achieved by dephosphorylation of these enzymes. Cyclic PIP regulates key metabolic pathways. For example, cyclic PIP activates up to 5-fold the pyruvate dehydrogenase complex. It inhibits the cyclic AMP-activated glycogen phosphorylase in liver and the cyclic AMP-stimulated lipolysis in adipocytes and it inhibits the adenylate cyclase [6].

This regulation of the equilibrium between the phospho- and de-phospho-form of an interconvertible enzyme is not recognized by all. One view is that metabolic regulation may be achieved by phosphorylation cascades alone [22]. It has also been more difficult to study the phospho-protein phosphatases, which are more labile enzymes than the kinases. Therefore, most studies have been done not with holoenzymes but with the catalytic subunits of these enzymes [23]. Separation of the regulatory and catalytic subunits caused a 10-fold increase in phosphatase activity, but precluded understanding of regulatory mechanisms by which this class of enzymes is activated, comparably to the activation of PKA by cyclic AMP. Protein ser/thr phosphatase holoenzyme was purified by ammonium sulfate precipitation, anion exchange chromatography (leading to two separate peaks of enzyme) and gel filtration [9]. It is not yet determined which one of the regulatory subunits, such as PP1 and PP2, are present in these holoenzyme preparations of protein ser/thr phosphatase. The inhibition of PKA by cyclic PIP is partly characterized: Cyclic PIP inhibits the PKA non-competitive binding to ATP and cyclic AMP; it inhibits the holoenzyme and the separated catalytic subunit; and its binding site on the PKA is near the cleft between the R- and C-subunits [6]. A comparable characterization of the activation of the protein ser/thr phosphatase has not yet been determined.

The characterization of cyclic PIP proceeded slowly in part due to the chemical labilities of this compound, bestowed by the 5-ring phosphodiester structure because of its ring tension, the allyl ether bond combining two secondary alcohols and the β-hydroxyketone structure of the prostaglandin E. In contrast to cyclic AMP, which is rapidly synthesized in a one-step reaction from ATP, the synthesis of cyclic PIP is complex. After hormonal stimulation, various inositol phosphates are liberated from lipid depots in the plasma membrane, of which inositol (1:2-cyclic,4)-bisphosphate is the substrate for the synthesis of activated inositol cyclic phosphate (Gypakis et al., manuscript to be submitted). The plasma membrane also stores unsaturated fatty acids for the synthesis of prostaglandins. Activated inositol cyclic phosphate and prostaglandin E are then combined by cyclic PIP synthase to cyclic PIP [20,21]. Presently, there are no arguments found why the synthesis of these two regulators is so different.

As discussed, cyclic PIP triggers the dephosphorylation of enzymes that were phosphorylated by cyclic AMP action. For further understanding of insulin action, it would be important to determine which insulin-activated protein kinases, such as PKB/Akt or MEK, are also activated by cyclic PIP. The bulk of current research concentrates on the characterization of kinase cascades by which the insulin signal is carried into the cells of a body [22]. This has led to the view that insulin simultaneously stimulates protein phosphorylation and dephosphorylation [24]. However, simultaneous phosphorylation and dephosphorylation reactions running in a cell at the same time could be very inefficient since cross-running, contradictory reactions may occur, unless 100% selectivity is provided in some other way such as differing intracellular locations.

Cyclic AMP acts by stimulating phosphorylation cascades. For instance, it activates PKA, which leads to the successive phosphorylation and activation of phosphorylase kinase and then glycogen phosphorylase. Two different mechanisms have been proposed for insulin’s action. The first one proclaims that insulin activates the insulin receptor tyrosine kinase, which then phosphorylates on tyrosine residues the insulin receptor substrates (IRSs), which are suggested to be docking proteins. This phosphorylation favors the binding of specific proteins/enzymes to the IRSs. In this way, the signal is spread into the cell [25]. However, presently it is not known how the regulatory effects of cyclic AMP could be reversed by this signal cascade. The second proposed mechanism invokes the increased synthesis of cyclic PIP which, as a small molecule, spreads rapidly in a cell, signaling protein dephosphorylation and reversing the action of cyclic AMP.

Known biological effects of cyclic PIP include the 10-fold activation of glucose uptake into adipocytes [26], the inhibition of glucagon-stimulated autophagy and proteolysis in isolated hepatocytes [27], the reduction of the binding of glucagon to its receptor [27], the inhibition of insulin-release from pancreatic β-cells within 2 to 3 min [6]. Cyclic PIP also triggers a 2.7-fold positive inotropic effect on the papillary muscle of the heart, which is associated with an elongation of the contraction time. This matches the inotropic effect evoked by noradrenalin (α-adrenergic receptor action) and also insulin stimulation, whereas the effect of cyclic AMP is associated with a shortening of the contraction time [27]. Furthermore, in studies of the slime mold *Dictyostelium disc.*, cyclic PIP antagonizes the effects of cyclic AMP to form fruiting bodies and sporulate. In this case, cyclic PIP may function as a signal that keeps the slime mold unicellular and growing [6]. Taken together, all these effects of cyclic PIP suggest that cyclic PIP is the intracellular regulator that triggers anabolism, just as cyclic AMP triggers catabolism. Cyclic AMP activates PKA, the cyclic AMP-activated guanine exchange factors and the cyclic nucleotide gated ion channels. The comparable discovery of the regulatory properties of cyclic PIP is still in progress, restricting this review predominantly to the regulation of phosphorylation and dephosphorylation of proteins/enzymes.

Diabetologists are focused on decreasing the elevated blood glucose levels of type 2 diabetics, based on the belief that increased protein glycation is a major cause of secondary health issues of diabetics [28]. It is a goal of diabetologists to maintain the blood glucose level at about 200 mg/dl in long term type 2 diabetics. Measurement of blood glucose levels is a rapid way to determine the metabolic state of a person. However, a better longtime goal should be to improve the insulin resistance of the type 2 diabetics, since the secondary health issues are more likely the result of a weakened signal-transduction of insulin, leading to inefficient functioning of anabolic pathways.

## 4. Stimulation of Cyclic PIP Synthesis by Metformin

All biguanides tested (metformin, buformin and phenformin) activated cyclic PIP synthase [10]. Metformin increased it by 1.9 to 2-fold; buformin 3.0 to 3.2-fold and phenformin 3.5 to 4.2-fold (Table 1). The most likely action of the biguanides is the activation of the insulin receptor tyrosine kinase [29], which then activates cyclic PIP synthase by tyrosine phosphorylation [21], leading to elevated cyclic PIP synthesis. Interestingly, this activation is additive to the activation of cyclic PIP synthase by fluoride [20,21], which was suggested to activate G proteins comparably to the activation of adenylate cyclase via G-proteins. Fluoride may also act as a protein tyrosine phosphatase inhibitor. In this case, fluoride inhibits the dephosphorylation of the, by tyrosine phosphorylation, activated cyclic PIP synthase [21]. However, the activation of basal cyclic PIP synthase by fluoride in the absence of ATP can best be explained by the action of a G protein (Table 1), suggesting that fluoride may exert two independent effects on cyclic PIP synthase.

The activation of cyclic PIP synthase by tyrosine phosphorylation [21] predicts that it is inactivated by dephosphorylation (Figure 2). Consequently, a further way to keep cyclic PIP synthase active is to inhibit this protein tyrosine phosphatase. The antihypertensive drug captopril was shown to inhibit protein tyrosine phosphatase [30] and it was shown that captopril and other antihypertensive drugs increase the synthesis of cyclic PIP [10]. This suggests that a more efficient treatment of type 2 diabetics should be to treat these patients with a combination of these 2 drugs or their alternatives.

**Figure 2 ijms-23-02200-f002:**
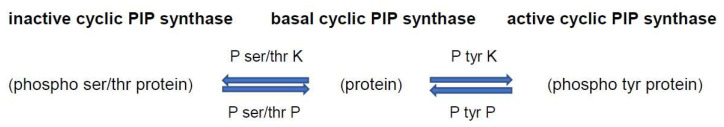
Regulation of the activity of cyclic PIP synthase by protein tyrosine and protein ser-ine/threonine phosphorylation and dephosphorylation. On insulin stimulation, the insulin receptor protein tyrosine kinase is activated and activates by tyrosine phosphorylation cyclic PIP synthase. Increased cyclic AMP levels activate P ser/thr kinases, which then inhibit cyclic PIP synthase. Abbreviations: P tyr K, protein tyrosine kinase; P tyr P, protein tyrosine phosphatase; P ser/thr K, protein serine/threonine kinase; P ser/thr P, protein serine/threonine phosphatase.

Past experience has shown that metformin loses its efficacy to help type 2 diabetics within five to ten years of treatment. A more recent observation is that metformin helps diabetics for a longer time [31]. Metformin appears to act intracellularly on the insulin receptor (Figure 3), activating down-stream signaling [32]. The progression of insulin resistance is connected to a decreasing receptor density [33,34]. Thus, the primary site of action of metformin and the location where effects of insulin resistance begin are closely located. It is of interest to ask whether the efficacy of metformin is affected by the decreasing receptor density. The two regulators, cyclic AMP and cyclic PIP, inhibit the synthesis and action of each other [6]. There are at least two ways by which cyclic PIP synthesis is inhibited. Metformin activates cyclic PIP synthesis very well, but it is not yet known whether it shuts off the inhibitory mechanisms, such as (a) inactivation of cyclic PIP synthase by protein tyrosine dephosphorylation [21] and (b) inhibition by protein serine/threonine phosphorylation, which is triggered by cyclic AMP [10]. Better understanding of the interplay of the inhibitory mechanisms on cyclic PIP synthase will further improve the treatment of diabetics.

A rare but serious side effect of the biguanides is that they can cause lactic acidosis [35,36]. A puzzling point is that the biguanides buformin and phenformin are more efficient in stimulating cyclic PIP synthase than metformin and one would expect that these biguanides would be more beneficiary for diabetics. It is rather peculiar that these more efficient biguanides are the ones with the greater risk to cause lactic acidosis. Could this problem of lactic acidosis be connected to overdosage of these drugs, especially as overdosing of insulin results in hypo-glycemia? The same problem is expected in cases of excess levels of cyclic PIP. Cyclic PIP stimulates the pyruvate dehydrogenase complex up to 5-fold [6]. This means that the intracellularly produced pyruvate should not be converted to lactate but to acetyl-CoA, and the question remains why is pyruvate transformed in high amounts to lactate. One possibility is that in these patients the biguanide could not increase cyclic PIP synthesis. This could result, for instance, from consumption of easily accessible, non-steroidal-anti-inflammatory drugs, which inhibit prostaglandin synthesis preventing cyclic PIP synthesis [26]. Furthermore, type 1 and type 2 diabetes have a tendency to develop lactic acidosis, and a large study showed an incidence rate of 3% for the patients with diabetes mellitus compared to an incidence rate of 0.1% in the non-diabetic control group [37]. Other studies came to the conclusion that the use of metformin did not increase the risk of lactic acidosis [38,39]. However, the well-recognized risk of lactic acidosis is feared because of its high mortality rate [40]. Though risk-factors for the development of lactic acidosis, such as hypoxia, sepsis, dehydration, and worsening of renal and cardiac failure, are recognized, more studies are needed to fully understand the problem at a molecular level and to further reduce/prevent the occurrence of lactic acidosis in type 2 diabetic patients on treatment with metformin. A question that needs to be answered is: Is the development of lactic acidosis more the result of other risk factors in the diabetic patient or does it solely result from the treatment with biguanides? All biguanides stimulate the synthesis of cyclic PIP, which activates pyruvate dehydrogenase, preventing the overproduction of lactate. Thus, it should not be a biguanide that initiates overproduction of lactate leading to lactic acidosis.

The goal of research should be to ask for a more effective drug than metformin. One can remember phenformin, which activates the biosynthesis of cyclic PIP twice as effectively as metformin, though phenformin was taken from the market because of the lactic acidosis problem. If causes of lactic acidosis become better understood and found not to be the result of the action of the biguanides, a drug such as phenformin might be reconsidered.

The regulation of the activity of cyclic PIP synthase appears to be modulated by many drugs. For instance, antidiabetic drugs of the sulfonylurea group inhibit cyclic PIP synthase (Table 1) [10]. These drugs are known to increase insulin secretion and one may ask if this can be beneficiary, especially when one takes into consideration that insulin and cyclic PIP switch off the secretion of insulin [6]. Further, it is known that in the early stage of type 2 diabetes patients have elevated serum insulin levels, most likely as a result of the reduced or lost switch off mechanism. Certainly, one can see the increase in insulin secretion as positive and helpful, but one needs to be aware that these drugs inhibit cyclic PIP synthase not only in the β-cells of the pancreas but in all cells of a body [10] and this inhibition will decrease the efficacy of insulin.

## 5. Conclusions

Many common illnesses such as type 2 diabetes, cancer, Alzheimer disease, and chronic inflammatory diseases, are connected to hyperphosphorylation. In diabetic and hypertensive rodents, the synthesis of cyclic PIP is decreased compared to the controls [10], favoring the dominance of the cyclic AMP signal. Such an issue was discussed years ago by Earl Sutherland [41]. An imbalance between these two regulators will lead to an ever-increasing imbalance. A dominance of the cyclic AMP signal will result in increased phosphorylation of proteins. Decreased synthesis of cyclic PIP will increase hyperphosphorylation because of a decreased or too slow dephosphorylation of phospho-proteins. Hyperphosphorylation of cellular proteins/enzymes is a point of concern nowadays. The two-fold stimulation of cyclic PIP synthesis by metformin will certainly help to reduce the hyperphosphorylation. However, the maximal effect of metformin is reached at a 5 × 10^−4^ molar concentration [10], and the concentration of metformin reached in the cells of diabetic patients is 4 × 10^−5^ molar [42]. At this concentration, cyclic PIP synthesis is increased by about 50% [10]. However, actual stimulation of cyclic PIP synthesis by metformin may be greater than expected because metformin concentration is constant over a prolonged time and this will continue to stimulate cyclic PIP synthesis, whereas cyclic PIP persists only briefly about 10 min after a hormonal stimulation.

Protein dephosphorylation could be the dominant action of metformin and this could explain why metformin shows beneficiary effects in many illnesses. But one has yet to determine optimal treatment protocols. Ideally, if metformin was 100% effective, type 2 diabetics should be cured without developing the many life-threatening illnesses such as diabetic foot, kidney failure, stroke, and blindness. The question is whether patients develop these problems because metformin’s stimulation of cyclic PIP synthesis is not strong enough to completely counter the action of cyclic AMP, whose levels are increased in diabetics [10]. It might be helpful to treat type 2 diabetic patients not only with metformin but with the addition of an adenylate cyclase inhibitor in order to bring the levels of cyclic AMP back into the normal range. This suggestion to inhibit the adenylate cyclase in order to support metformin’s action is backed by the effects that diabetic rodents have increased cyclic AMP levels [10] and the beta blockers nebivolol and carvedilol improve insulin sensitivity and glucose tolerance [43]. Furthermore, increased levels of adenylate cyclase 8 and thus increased cyclic AMP levels are associated with obesity and type 2 diabetes [44]. In addition, increased blood glucose levels (hyperglycemia and diabetes) cause an increase of cyclic AMP by adenylate cyclase 5 in arterial myocytes, resulting in activation of L-type Ca^2+^ channels and vasoconstriction [45].

In summary, metformin is a drug with many positive effects, which appear to result from its stimulation of the synthesis of cyclic PIP. However, it does not function at 100% efficacy at least in the case of treatment of diabetics. This handicap seems to be connected to the development of insulin resistance. This handicap may not be a problem in the treatment of patients with other illnesses, in which metformin is presently used with a positive outcome.

Finally, what are the goals for future studies of cyclic PIP? The primary goal is the chemical synthesis of cyclic PIP and the synthesized product must have the same regulatory properties as the compound isolated from rat liver. The chemical synthesis is difficult, since cyclic PIP is naturally difficult to study because of its lability and the chemical synthesis requires harsher conditions than biochemical isolation. Once pure cyclic PIP is available, experiments with various isolated cells and enzymes will need to be performed. For example, it will be of interest to determine if PKB/Akt, MEK and casein kinase I are activated by cyclic PIP [6]. It is also necessary to determine in which way cyclic PIP inhibits adenylate cyclase and if G proteins or hormone receptors could be involved. In addition, the effect of cyclic PIP for instance on the heart has to be characterized in more detail. Another key goal will be the further characterization of cyclic PIP synthase. It has to be unequivocally determined whether cyclic PIP synthase and the IRS proteins are identical or not. Presently, the IRS proteins are solely seen as docking proteins. Cyclic PIP synthase is activated on insulin stimulation by tyrosine phosphorylation [21], and anti-IRS-antibodies bind solubilized and gel filtrated cyclic PIP synthase [unpublished results]. The regulation of cyclic PIP synthase activity by protein ser/thr phosphorylation/dephosphorylation and protein tyr phosphorylation/dephosphorylation must be characterized in detail in healthy and type 2 diabetic patients. Overall, these approaches should lead to understanding of mechanisms that lead to insulin resistance.

## Figures and Tables

**Figure 1 ijms-23-02200-f001:**
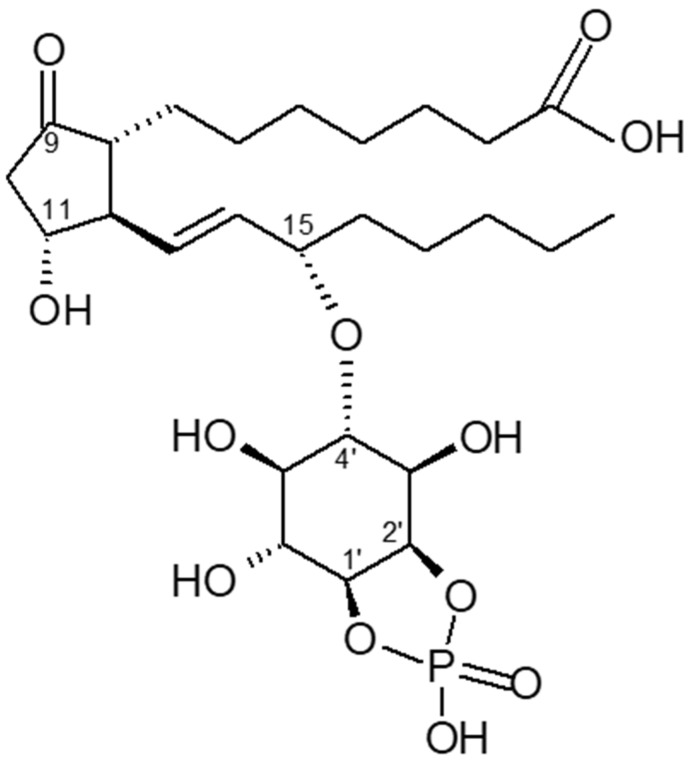
Proposed chemical structure of prostaglandylinositol cyclic phosphate (cyclic PIP). Inositol 1:2-cyclic phosphate is bound via its C4 hydroxyl group to the C15 hydroxyl group of prostaglandin E (PGE) (adapted from [18]).

**Figure 3 ijms-23-02200-f003:**
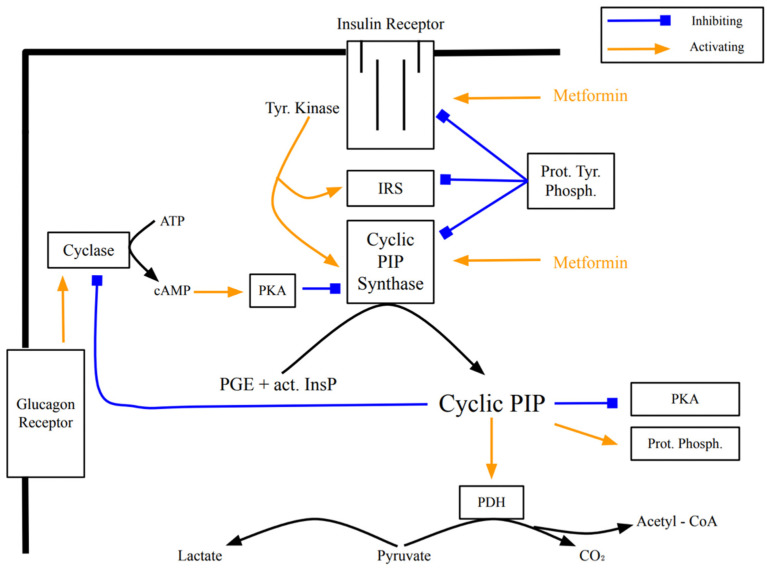
Signal transduction of insulin and the activation of cyclic PIP synthase by tyrosine phosphorylation, and regulatory effects of cyclic PIP. The points of action of metformin are indicated. (used abbreviations: Tyr Kinase is the insulin receptor tyrosine kinase; Prot Phosph the protein ser/thr phosphatase; act InsP the activated inositol cyclic phosphate; Cyclase the adenylate cyclase; Prot Tyr Phosph a protein tyr phosphatase; PDH the pyruvate dehydrogenase and PKA protein kinase A).

**Table 1 ijms-23-02200-t001:** Activation of particular cyclic PIP synthase by various chemicals and drugs.

	Cyclic PIP Synthase Activity	
Addition	Activation (x-fold)	Inhibition (%)
**Experiment 1**		
No Additions (basal activity)	1.0	
Fluoride (10mM)	3.2	
ATP (10mM)	10.6	
ATP + Fluoride	14.4	
**Experiment 2**		
Metformin (0.5mM)	2.1	
Buformin (1.0 mM)	3.05	
Phenformin (1.0 mM)	4.2	
Glibenclamide (0.1 mM)		66
(1.0 mM)		99
Chloropropamide (1.0 mM)		95
Tolbutamide (1.0 mM)		97

In experiment 1, the basal activity of cyclic PIP synthase was determined in the absence of ATP and fluoride in the assay; in experiment 2, the basal assay contained 10 mM ATP; (Results are adapted from [10,20,21]).

## Data Availability

Not applicable.

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
