# Peer review of "Metformin’s Mechanism of Action Is Stimulation of the Biosynthesis of the Natural Cyclic AMP Antagonist Prostaglandylinositol Cyclic Phosphate (Cyclic PIP)â€"

_ijms, 2022, doi:10.3390/ijms23042200_

Round 1
Reviewer 1 Report
This is an interesting review article describing cyclic PIP as a possible natural cAMP antagonist in the context of metformin's molecular mechanism of action in T2D. Overall I liked this manuscript and the discussion about this exciting signaling pathway, the concept of balancing excessive protein phosphoprylation and its potential as a drug target. There are some points where the paper can be improved:
- The field of cyclic PIP seems to be relatively small. About the first third of references are mostly review papers by the author himself. It would be nice to include here as many original research article references as possible, also from other groups, if available. Obviously, a lot of further research needs to be done- I would be happy to see some outlook/further perspectives at the end of the paper which could discuss such issues.
- In the abstract is is stated quite "bold" that the mechanism of metformin action os unknown. As far as I remember my pharmacology lessons etc, there have be some mechanisms proposed and there could be indeed several mechanisms which might explain the pleiotropic action of metformin. There have been some research e.g. on AMPK. Maybe mention what has been proposed and that the exact mechanisms is incompletely understood.
- As a reader I was interested what could be the molecular identity of cyclic PIP synthase- is the exact protein und e-g- human/mouse gene encoding it known? I found that "IRS-1-like enzyme" was mentioned by some other authors in connection with cyclic PIP. Please clarify. And what is the exact/possible binding domain of Cyclic PIP? I understood it can bind to and regulate the activity of phophatases, are those both PP1 and/or PP2A, is it known where the cyclic PIP binding domain is located in their sequences?
- at the beginning of section3 it is mentioned that cyclic PIP can be produced in many tissues by alpha-adrenergic and ins stimulation. Are there also cell lines which can synthetise it upon given stimulus and which could be potentially used as a model system. Alpha adrenergic means that the downstream G-protein is probably Gi. It is the case? How can one distinguish between the effects of cyclic PIP on phosphorylated proteins vs potential cAMP lowering effect via Gi, at least upon noradrenalin stimulation?
- greek letters alpha and beta are written as a and b thoughout the ms, please correct
- Figure 1 legend cites a paper at the end- is it Ref 9, maybe include the ref number
- Data in table 1 seem to have been taken from a publication, please cite it in the fig legend. I could find somewhat similar data in the ref 12 by the authors... please state the concentrations of buformin and phenphormin used.
- at the end of page 5 original references about the action of cyclic PIP on heart muscle are missing. instead it just cites a book chapter 18. please add.
- ref 11 is a manuscript in preparation which is supposedly non citable at least if not published online as a preprint...
bottom line- very interesting, I liked reading it. however, the above mentioned improvements would greatly help understanding and aid the manuscript.
Author Response
Concerning the manuscript ijms 1546102
Dear Referee 1,
I thank you for the comments and suggestions to improve the manuscript and respond as follows. The added and changed text parts are marked yellow, this works best with my typing skills, and I hope that this works well.
To point 1: further literature citations on cyclic PIP are given, see reference 7,8,9 and 19. An outlook on how the project will proceed is given at the end of the text file on page 9, lines 350-366 of the manuscript.
To point 2: on page 1 the lines 13 and on page 2 the lines 83-87 show the made changes on the mechanism of action of metformin. A further literature [14] is added.
To point 3: on page 9 the lines 358-363. It is informed on the extent of the characterization of cyclic PIP synthase. The goal is to show – what is most likely the case – that IRS and cyclic PIP synthase are identical. The IRSs are well characterized. When the identity can be proven, then a lot is known on cyclic PIP synthase. In case this will not be the case, the characterization will need time.
On page 4/5 the lines 149-157 answer what is known on the regulation of protein ser/thr phosphatase and protein kinase A with respect to their activity regulation by cyclic PIP.
Years ago, we had prepared pure PKA and started to characterize the inhibition of the PKA by cyclic PIP. Being able to bind photoaffinity-labeled cyclic PIP to the PKA, we could have shown at which part of the amino-acid-sequence of this enzyme cyclic PIP is bound on the R- and C-subunit of the PKA (having to move to another location this research could not be finished). Performing comparable experiments with the protein ser/thr phosphatase is/will be much more difficult because of the greater lability of this enzyme and the resulting difficulties to obtain a pure enzyme. Because of these difficulties scientists decided not to purify the holoenzymes but to separate the subunits of this class of enzymes and purify alone the regulatory subunits as for instance PP1 and PP2 (Dr. Brautigan in the cited review describes this).
To point 4: So far cyclic PIP has not been determined in a cell line grown in cell culture. However, from endothelial cells grown out from rat aortas, the inositol phosphate precursor for the synthesis of the activated inositol phosphate has been isolated, and it is very likely that these cells synthesize also a fair amount of cyclic PIP. But such experiments have not been performed.
The inhibition of adenylate cyclase by cyclic PIP has not been characterized any further. Adenylate cyclase is inhibited by cyclic PIP dose-dependently up to 100%. The inhibitory effect of the Gi protein is comparably a week effect (discussion with Dr. Thomas Pfeuffer, who was the first to define G proteins). A further goal to be reached is to characterize the inhibition of adenylate cyclase by cyclic PIP. This is written on page 9 in the lines 357-358.
To the distinction of the effect of Gi proteins and the inhibition by cyclic PIP: By Gi proteins the synthesis of cyclic AMP is reduced. By cyclic PIP is not only the synthesis of cyclic AMP inhibited, but the existent phosphorylation of proteins reversed by the action of the cyclic PIP-activated protein ser/thr phosphatase.
To point 5: I am sorry for this mistake. The original manuscript had Greek lettering, most likely on editing the manuscript and changing the letter-type, the Greek letters were changed. They are now back in Greek letters.
To point 6: The reference number has been added. It is [18].
To point 7: The applied concentrations (1 mM) of buformin and phenformin have been added in the Table 1, and the publications [10,18,21] from which the data have been taken has been added in the legend of the Table.
To point 8: I am sorry that these results are presently only shown in a book form. (A reason for this (though I dislike, to talk about it) is that I was forced by the decision of a German senior scientist that I get only the allowance to publish the chemical structure of cyclic PIP but anything else will not obtain his support.)
To point 9: The reference 11 has been erased.
I hope that the made corrections meet your expectations. I thank once more for the evaluation of this manuscript and the suggestions. I am feeling well because of the improvement of the manuscript.
Kind regards,
Heinrich Wasner
Reviewer 2 Report
This review aims to ellucidate the molecular mechanisms of action of metformin, specifically its interplay with the cAMP/PIP pathways. The manuscript is scientifically sound, logically interconnected and reads well.
I have only some small suggestions for an improvement of particularly the visual aspect of the paper:
- Please remove the citation from the Abstract section
- I am missing some references in the Introduction (lines 24-63) addressing the description of diabetes as well as providing evidence to the observation that DM2 patients may or may not be obese.
- A figure or scheme providing a comprehensive description of the behavior of metformin in stimulating the synthesis of cyclic PIP and thus inhibiting cAMP and modulating molecular responses requested by insulin would add a great visual bonus for the readers.
Author Response
Concerning manuscript ijms 1546102
Dear referee 2,
I thank you for the comments and suggestions to improve the manuscript and respond as follows. The added and changed text parts are marked yellow, this works best with my typing skills. I hope that this works well.
To point 1: The literature citation in the abstract has been removed.
To point 2: 5 literature citations are added now in the introduction, see the references [1] to [5].
To point 3: the figure 3 has been removed from the manuscript and a new figure 3 is added, which shows more details with respect to the discussed data.
I hope that the made corrections meet your expectations. I thank you once more for the evaluation of this manuscript and the made suggestions. I am thankful because the manuscript has been improved.
Kind regards,
Heinrich Wasner
Round 2
Reviewer 1 Report
The author has addressed my comments satisfactorily